# Operational indicators for pulmonary tuberculosis diagnosis in people living with HIV before and after Xpert MTB/RIF implementation in the state of São Paulo, Brazil

**Mariana Gaspar Botelho Funari de Faria**[1], **Rubia Laine de Paula Andrade**[1]*, **Karina Fonseca de Sousa Leite**[1], **Rafaele Oliveira Bonfim**[1], **Ana Beatriz Marques Valença**[1], **Antônio Carlos Vieira Ramos**[1], **Thais Zamboni Berra**[1], **Ricardo Alexandre Arcêncio**[1], **Maria Josefa Perón Rujula**[2], **Jaqueline Garcia de Almeida Ballestero**[1], **Erica Chimara**[3], **Antônio Ruffino Netto**[4], **Dulce Maria de Oliveira Gomes**[5], **Valdes Roberto Bollela**[4], **Aline Aparecida Monroe**[1]

1 Ribeirão Preto College of Nursing, University of São Paulo, Ribeirão Preto, São Paulo, Brazil, 2 Center of Epidemiological Surveillance at Sao Paulo State Department of Health, São Paulo, Brazil, 3 Adolfo Lutz Institute, Government of the State of São Paulo, São Paulo, Brazil, 4 Ribeirão Preto School of Medicine, University of São Paulo, Ribeirão Preto, São Paulo, Brazil, 5 School of Science and Technology, University of Évora, Évora, Alentejo, Portugal

* rubia@eerp.usp.br

## Abstract

Tuberculosis (TB) in people living with HIV (PLHIV) is usually paucibacillary and the smear microscopy has limitations and may lead to high proportions of non-confirmed pulmonary tuberculosis (NC-PTB). Despite culture being the reference method, it usually takes 6 to 8 weeks to produce the results. This study aimed to analyze the effect of a rapid molecular test (Xpert) in the confirmatory rate of PTB among PLHIV, from 2010 to 2020, in São Paulo state, Brazil. This is an ecological study with time series analysis of the trend and the NC-PTB rates before and after Xpert implementation in 21 municipalities. The use of Xpert started and gradually increased after 2014, while the rate of NC-PTB in PLHIV decreased over this time, being more significant between late 2015 and mid-2017. The city of Ribeirão Preto stands out for having the highest percentage (75.0%) of Xpert testing among PLHIV and for showing two reductions in the NC-PTB rate. The cities with low Xpert coverage had a slower and smaller decrease in the NC-PTB rate. Despite being available since 2014, a significant proportion of PLHIV suspected of PTB in the state of São Paulo did not have an Xpert ordered by the doctors. The implementation of Xpert reduced the NC-PTB rates with growing effect as the coverage increased in the municipality.

## Introduction

Tuberculosis (TB) has been considered a global emergency by the World Health Organization (WHO) since the 1990s [1]. A total of 7.1 million TB cases were reported worldwide in 2019;

**Data Availability Statement:** All relevant data are within the manuscript and its Supporting Information files.

**Funding:** This study was financed by the Coordenação de Aperfeiçoamento de Pessoal de Nível Superior – Brasil (CAPES) [Financial Code 001], Conselho Nacional de Desenvolvimento Científico e Tecnológico (CNPq) – Produtivity in Research Sponsorship [grant number 317170-2021-0] and Fundação de Amparo à Pesquisa do Estado de São Paulo [process number 2022/00025-2].

**Competing interests:** The authors have declared that no competing interests exist.

however, this number decreased to 5.8 million in 2020 due to the COVID-19 pandemic [2]. The following year, 6.4 million TB cases were reported globally, with 6.7% co-infected with HIV, and 1.6 million resulting in death, including 187,000 deaths in people living with HIV (PLHIV) [2].

Brazil reported 68,271 new TB cases in 2021 [3], being among the 30 prioritized countries for controlling TB and TB/HIV co-infection [2]. Approximately 23% of the cases in the country are concentrated in the state of São Paulo [4].

PLHIV have 15 to 21 times higher chances of developing active TB after infection compared to the general population [5]. Moreover, 8.3% of reported TB cases in Brazil in 2021 were associated with HIV co-infection [3]. Given this scenario and in the context of achieving global goals for TB elimination as a public health problem, there is a reaffirmed need to improve case detection, as it is a crucial element in reducing TB transmission [3].

In this context, TB diagnosis is based on clinical and radiological findings, confirmed through laboratory tests such as smear microscopy and culture [6]. Despite being used for decades, these methods have limitations: low sensitivity (smear microscopy) and delayed results (culture). Despite culture being the reference test for TB diagnosis, it is seldom used for decision-making related to TB diagnosis and treatment in most situations [6, 7].

Thus, the World Health Organization approved the use of the GeneXpert® system for rapid molecular testing of TB (RMT-TB) in the mid-2010s to overcome these limitations. This molecular method is based on polymerase chain reaction (PCR), and can also detect the most common mutations which confers resistance to rifampicin, a key antibiotic in TB treatment [8, 9].

Unlike other real-time PCR assays, Xpert is a simple, automated technique that requires little training, providing rapid accurate and reliable results [10, 11]. Moreover, its use involves substantially lower biological risks compared to smear microscopy, as manual preparation is simple with the sample being inserted into a cartridge containing a bactericidal buffer [12].

Despite advancements with the implementation of this new diagnostic test, failures may occur in case confirmation, potentially impacting prognosis, case mortality, transmission of drug-resistant strains, and inappropriate treatment initiation [13]. Addressing such shortcomings poses a challenge to health services, especially among PLHIV, where atypical presentations of PTB and treatment without laboratory confirmation is still a challenge [14].

In this context, studies indicate some promising expectations regarding the rapid molecular test, such as increased case detection and timely TB treatment, contributing to a reduction in incidence and mortality from the disease [15, 16]. Regarding increased case discovery, a literature review in December 2019 highlighted higher accuracy of Xpert in confirming pulmonary TB (PTB) in PLHIV compared to smear microscopy, and showed similar performance to culture and much faster [17]. Many studies in this review assessed its accuracy with measures such as sensitivity, specificity, positive and negative predictive values, among others, so the present study will analyze it through an operational indicator.

In view of the above, this study aims to analyze the non-confirmed PTB (NC-PTB) diagnosis before and after Xpert implementation for PLHIV in municipalities at the state of São Paulo, Brazil, from 2010 to 2020.

## Methods

This is an ecological time series (TS) study [18], characterized as a set of observations obtained sequentially over time. Data representation in the temporal domain is of great importance, as the relationship between chronologically adjacent observations reflects the interdependence that one observation maintains with another [19]. This study was developed based on the

Strengthening the Reporting of Observational Studies in Epidemiology (STROBE) initiative checklist [20] and the suggested additions to the STROBE document for ecological reporting [21].

The study was conducted in some municipalities in the state of São Paulo, which is in the Southeast region of Brazil. The following 21 municipalities were considered in the study because they implemented Xpert® MTB/RIF for PTB diagnosis in 2014 and 2015, and so they constituted the observation units of the study: Araçatuba, Bauru, Barueri, Bragança Paulista, Campinas, Carapicuíba, Franco da Rocha, Guarulhos, Itapecerica da Serra, Jundiaí, Marília, Presidente Prudente, Ribeirão Preto, Santo André, Santos, São Bernardo do Campo, São José do Rio Preto, São Paulo, Sorocaba, Taubaté, and Tremembé. All municipalities have at least one laboratory to conduct the tests, which were processed by the Cepheid GeneXpert® System. The initially implemented Xpert® MTB/RIF cartridges were completely replaced by Xpert® Ultra in the last quarter of 2019.

Data from the Tuberculosis Patient Control System (TB-WEB) regarding new cases of pulmonary TB with HIV coinfection diagnosed from 2010 to 2020 in the mentioned municipalities were collected. The following variables were used to identify new cases of pulmonary TB with HIV coinfection: case type (to identify new TB cases), clinical form (to identify cases of pulmonary TB); HIV, AIDS, and antiretroviral therapy (to identify HIV infection). Individuals under 18 years old and those with diagnostic change were excluded.

Anonymous data from TB-WEB were made available in October 2021 by the State Epidemiological Surveillance Center and included the following variables: identification data (notification municipality, diagnosis date), sociodemographic data (age, sex, race/color, education), comorbidities (none, diabetes mellitus, mental disorders, AIDS, other immunodeficiency), use of psychoactive substances (smoking, alcoholism, illicit drug use), and tests performed and results (RMT-TB, sputum smear microscopy, sputum culture, sensitivity test, and X-ray).

After data collection, all study variables were analyzed using descriptive statistics (absolute and relative frequency distribution for qualitative variables and measures of position and variability for quantitative variables). The Chi-square and Fisher's exact tests were applied to compare patients characteristics and tests undergoing in the pre- and Post-Xpert implementation periods.

The NC-PTB rates were subsequently calculated monthly for the state and quarterly for each municipality. The molecular testing realization rate among PLHIV and PTB was additionally calculated for the state and each studied municipality. It is important to note that data from the 21 municipalities considered for the study were aggregated for calculations involving the state. All of these calculations were performed in Excel spreadsheets, considering:

$$X_i = \frac{\text{new cases of pulmonary TB among individuals with type I HIV}}{\text{new cases of pulmonary TB with HIV coinfection}} \times 100, i = 1, 2$$

In which: 1—not confirmed etiological diagnosis; 2—RMT-TB performance.

The time series rates of the state and municipalities were then decomposed through the formula: ($Y_t = S_t + T_t + R_t$, t = 1,. . .,N), where the monthly/quarterly rates are expressed by $Y_t$, the seasonal component by $S_t$, the trend component by $T_t$, and finally the residual or noise component by $R_t$. An additive decomposition was chosen, as the variabilities of the observations tend to not increase over time.

The Seasonal Decomposition of Time Series by LOESS (STL) method was used from the perspective of estimating each component of the time series [22].

The trend refers to the direction in which the time series develops over a period, which can be increasing or decreasing, with the possibility of being linear or non-linear. Seasonality is

defined by identical patterns that repeat periodically and regularly in fixed time periods. The noise represents the observed fluctuations during the series period, usually irregular and random, and only noticeable when the other components of the time series are removed [23].

A structural change analysis was subsequently performed on the time series. In this analysis, we determined if there was any structural break in the time series, and if so, at what point it occurred with their respective confidence intervals using the *strucchange* package in R version 4.2.2. Zeileis et al. (2022) [24] provide theoretical details and examples of the effectiveness of this approach.

It is worth noting that due to some municipalities not reporting any cases in the analyzed quarters, it was only possible to analyze the time series of the NC-PTB rates in eight municipalities. Therefore, a descriptive analysis of these rates was carried out in the periods before and after implementing RMT-TB for the analysis of the remaining 13 municipalities, comparing the rates in the pre- and post-Xpert implementation periods using the Chi-square and Fisher's exact tests.

The present study is in compliance with Resolution 499/2012 of the National Health Council, and was approved by the Research Ethics Committee of the Ribeirão Preto Nursing School at the University of São Paulo, under opinion number 4,478,072. The participants did not have to assign an Informed Consent Form as the data were anonymous and collected through a secondary source.

## Results

A total of 193,094 new PTB cases were reported in the state of São Paulo from 2010 to 2020. Out of these cases, 174,283 had excluded HIV infection, 4,308 had extrapulmonary TB, 728 had a change in the final diagnosis, 260 were under 18 years old, and 4,082 cases were reported in municipalities that did not implement Xpert. All of these cases were excluded.

The remaining 9,433 cases were classified as PTB in PLHIV reported in municipalities that implemented Xpert. There were 4,035 (42.8%) cases before implementing Xpert, and 5,398 (57.2%) after.

Table 1 shows that the TBP patients' ages varied from 31 to 50 years in 59.1%, and the mean ages before and after Xpert implementation were 39.1 years (SD: 10.4) and 39.9 years (SD: 12.7), respectively. Males predominated (74.8%), with white (47.1%) and brown (37.9%) race/ color accounting for 85% of the total cases. The majority of participants had 4 to 11 years of education (77.5%), and 21.1% reported use of illicit drugs, 17.6% alcoholism, and 13.2% smoking. AIDS manifested in 84.8% of cases. Patients' characteristics differ in most variables when they were compared in the pre- and post-Xpert implementation periods, except for sex and mental disorders.

The percentages of individuals with confirmed diagnosis for PTB in the period before Xpert implementation were as follows: smear microscopy, 87.9%; culture, 59.0%; and either of the two tests, in 89.2%. Then, the percentages of confirmed PTB diagnosis after implementation were: Xpert, 49.4%; smear microscopy, 69.2%; sputum culture, 56.1%; any of the three tests, in 88.1%. Despite being available, not all PTB cases had a molecular test ordered and performed and the coverage of the exam varied among municipalities.

Fig 1 illustrates that the coverage of Xpert gradually increased after 2014 in the state (represented by the pink line) and the NC-PTB diagnosis rate (green line) showed a decreasing trend during the study period, and more intense between late 2015 and mid-2017.

Table 2 shows that the municipality of Bragança Paulista with the lowest coverage of Xpert (12.0%), while Ribeirão Preto had the highest rate (75.0%). If we consider any of the three tests available to confirm PTB, the municipality of Bauru had the lowest percentage (76.3%) in the

**Table 1. Sociodemographic profile, comorbidities and use of psychoactive substances of cases of pulmonary tuberculosis and HIV coinfection in the pre- and post-Xpert implementation periods in 21 municipalities in the state of São Paulo, Brazil, from 2010 to 2020.**

| Variables | | Pre-Xpert implementation n (%) | Post-Xpert implementation n(%) | Total n(%) | p* |
|---|---|---|---|---|---|
| **Age group (years) N = 9,409** | 18–30 | 889(22.0) | 1,361(25.3) | 2,250(23.9) | <0.001 |
| | 31–40 | 1,393(34.5) | 1,675(31.2) | 3,068(32.6) | |
| | 41–50 | 1,187(29.4) | 1,303(24.2) | 2,490(26.5) | |
| | 51–60 | 453(11.2) | 680(12.6) | 1,133(12.0) | |
| | 61–70 | 92(2.3) | 247(4.6) | 339(3.6) | |
| | 71–96 | 19(0.5) | 110(2.0) | 129(1.4) | |
| **Sex N = 9,433** | Male | 2,982(73.9) | 4,077(75.5) | 7,059(74.8) | 0.072 |
| | Female | 1,053(26.1) | 1,321(24.5) | 2,374(25.2) | |
| **Education N = 6,655** | None | 74(2.5) | 81(2.2) | 155(2.3) | <0.001 |
| | 1–3 years | 281(9.3) | 275(7.6) | 556(8.4) | |
| | 4–7 years | 1,167(38.7) | 1,206(33.1) | 2,373(35.7) | |
| | 8–11 years | 1,200(39.8) | 1,583(43.5) | 2,783(41.8) | |
| | 12–14 years | 196(6.5) | 342(9.4) | 538(8.1) | |
| | 15 years or more | 97(3.2) | 153(4.2) | 250(3.8) | |
| **Race/ethnicity N = 8,439** | Asian | 19(0.5) | 21(0.4) | 40(0.5) | <0.001 |
| | Black | 490(13.6) | 720(14.9) | 1,210(14.3) | |
| | Brown** | 1,229(34.1) | 1,968(40.7) | 3,197(37.9) | |
| | Indigenous | 9(0.2) | 5(0.1) | 14(0.2) | |
| | White | 1,859(51.6) | 2,119(43.8) | 3,978(47.1) | |
| **Comorbidities*** | None | 117(2.9) | 227(4.2) | 344(3.6) | <0.001 |
| | Diabetes Mellitus | 72(1.8) | 173(3.2) | 245(2.6) | <0.001 |
| | Mental disorder | 59(1.5) | 94(1.7) | 153(1.6) | 0.288 |
| | AIDS | 3,671(91.0) | 4,332(80.3) | 8,003(84.8) | <0.001 |
| | Other Immunodeficiency | 18(0.4) | 55(1.0) | 73(0.8) | 0.002 |
| **Use of psychoactive substances*** | Smoking | 148(3.7) | 1,101(20.4) | 1,249(13.2) | <0.001 |
| | Alcoholism | 590(14.6) | 1,071(19.8) | 1,661(17.6) | <0.001 |
| | Illicit Drug Use | 712(17.6) | 1,278(23.7) | 1,990(21.1) | <0.001 |

"N" (number) may vary due to ignored or blank data found in the database.

*p-value for Chi-square test

** This is a Brazilian classification

***More than one option could be checked.

period after implementing Xpert, while Araçatuba and Jundiaí tested all the cases (100%) of PTB in PLHIV with one of the three available tests (smear microscopy, culture or Xpert) during this period. Only the municipality of Sao Paulo had a light difference in performing at least one test available when comparing the pre- and post-Xpert implementation periods.

The municipalities of Bauru and Guarulhos showed an increasing trend in NC-PTB diagnosis, but with a very irregular performance over the period, fluctuating with both increases and decreases in the rate. The municipalities of Santos and Santo André did not show a decreasing trend, even after implementing Xpert (Fig 2).

The city of São Paulo showed a decreasing trend in NC-PTB. One before the implementation of Xpert, and another in 2019, showing the complementarity of these diagnostic strategies (Fig 3).

The municipalities of São José do Rio Preto, Campinas, and Ribeirão Preto showed a decreasing trend in NC-PTB and it seems to be quite related to the implementation of Xpert

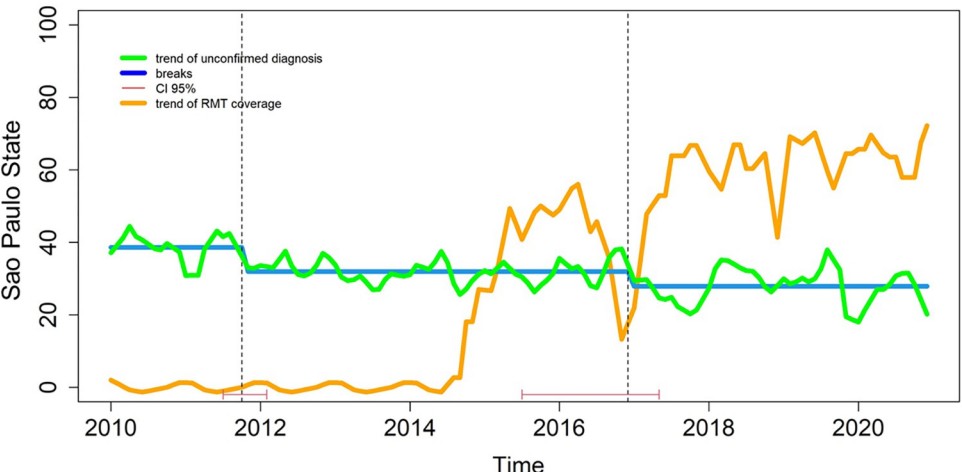

**Fig 1. Temporal trend of non-confirmed pulmonary tuberculosis and the coverage of rapid molecular test (Xpert) among patients with TB/HIV coinfection in 21 municipalities in the state of São Paulo between 2010 and 2020.**

**Table 2. Xpert coverage or at least one of the three available test for pulmonary tuberculosis in people with TB/HIV coinfection from 2010 to 2020 reported in 21 municipalities in the state of São Paulo, Brazil.**

| Municipality | Xpert after implementation | At least one test* in pre-Xpert implementation period | At least one test** in post-Xpert implementation period | p |
|---|---|---|---|---|
| | n/N(%) | n/N(%) | n/N(%) | |
| Bragança Paulista | 3/25(12.0) | 14/14(100.0) | 23/25(92.0) | 0.405*** |
| Itapecerica da Serra | 5/32(15.6) | 11/12(91.7) | 31/32(96.9) | 0.476*** |
| Franco da Rocha | 4/18(22.2) | 14/15(93.3) | 17/18(94.4) | 0.710*** |
| Bauru | 31/97(32.0) | 61/79(77.2) | 74/97(76.3) | 0.885**** |
| Presidente Prudente | 16/45(35.6) | 35/45(77.8) | 35/45(77.8) | - |
| Barueri | 14/38(36.8) | 31/32(96.9) | 34/38(89.5) | 0.237*** |
| Guarulhos | 69/162(42.6) | 95/98(96.9) | 148/162(91.4) | 0.078**** |
| Araçatuba | 7/16(43.8) | 5/5(100) | 16/16(100) | - |
| São José do Rio Preto | 44/99(44.4) | 94/111(87.7) | 88/99(88.9) | 0.371**** |
| Carapicuíba | 17/37(45.9) | 27/28(96.4) | 31/37(83.8) | 0.108*** |
| Marília | 30/63(47.6) | 49/49(100) | 61/63(96.8) | 0.314*** |
| São Paulo | 1,832/3,792(48.3) | 2,385/2,683(88.9) | 3,299/3,792(87.0) | 0.022**** |
| Sorocaba | 54/109(49.5) | 79/92(85.9) | 96/109(88.1) | 0.643**** |
| Taubaté | 28/54(51.9) | 31/38(81.6) | 48/54(88.9) | 0.322**** |
| São Bernardo do Campo | 43/82(52.4) | 56/60(93.3) | 75/82(91.5) | 0.468*** |
| Santo André | 63/109(57.8) | 86/93(92.5) | 102/109(93.6) | 0.758**** |
| Santos | 113/187(60.4) | 150/175(85.7) | 171/187(91.4) | 0.086**** |
| Campinas | 123/198(62.1) | 159/169(94.1) | 185/198(93.4) | 0.798**** |
| Jundiaí | 29/46(63.0) | 37/40(92.5) | 46/46(100) | 0.096*** |
| Tremembé | 14/21(66.7) | 9/9(100) | 19/21(90.5) | 0.483*** |
| Ribeirão Preto | 126/168(75.0) | 173/188(92.0) | 159/168(94.6) | 0.325**** |

*Smear microscopy or Culture

**Xpert or Smear microscopy or Culture

***p-value for Fisher's exact test

****p-value for Chi-squared test

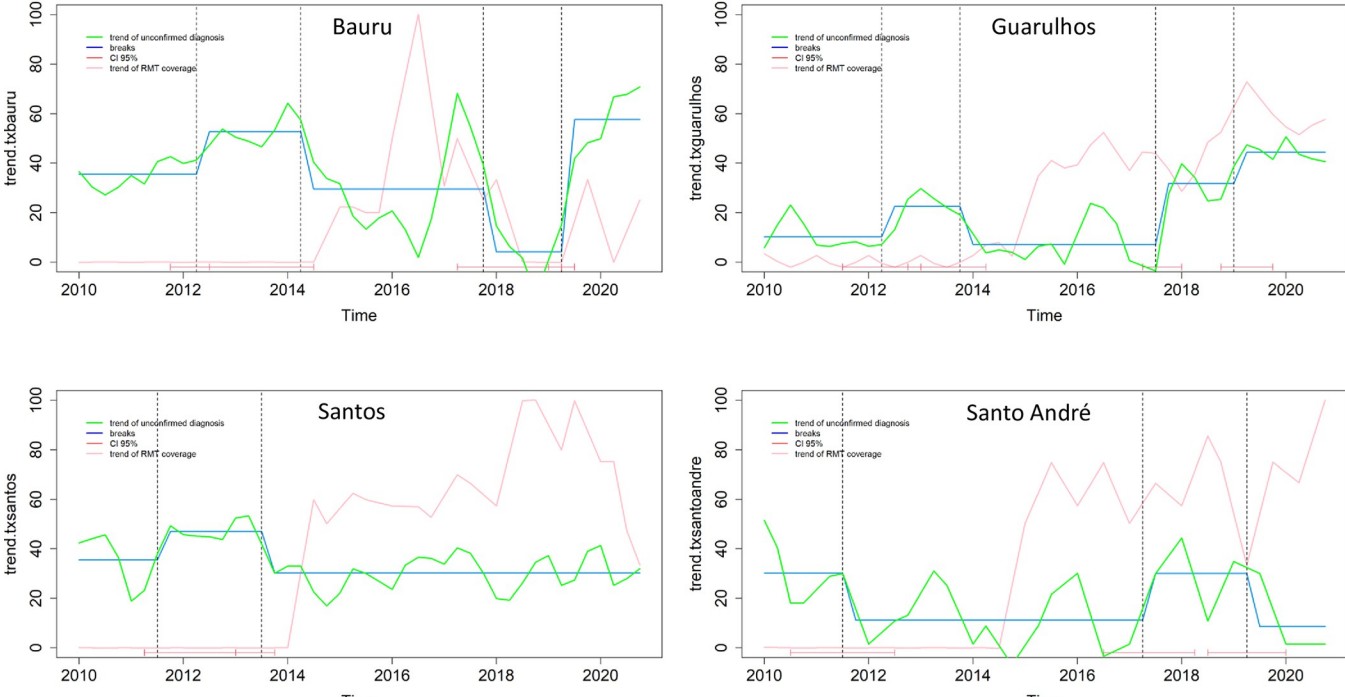

**Fig 2. Breaks and temporal trend of non-confirmed pulmonary tuberculosis among people with TB/HIV coinfection reported in four municipalities in the state of São Paulo between 2010 and 2020.**

(as per the presented confidence intervals—red lines at the bottom of the figures). Among these municipalities, Ribeirão Preto stands out for the increasing trend in Xpert coverage over the period and for showing a significant decrease in NC-PTB at the end of the study period (Fig 3).

In Table 3, it can be observed that the NC-PTB rates in PLHIV decreased in the municipalities of Bragança Paulista, Itapecerica da Serra, Jundiaí, and Taubaté.

## Discussion

The sociodemographic profile of the cases included in this study reinforces the association of the disease with social issues and implications, as there is a predominance of cases in economically active individuals, males, and those with 4 to 11 years of education. There is also a predominance of white race/ethnicity among the cases of PTB in PLHIV, followed by the "Brown" population. This scenario may be due to the higher prevalence of these races/ethnicities in the Southeast region of Brazil which is home to 49.9% of people who self-identify as White, and 37.6% as "Brown" [25, 26].

Regarding the comorbidities presented by the study population, there is a negligible number of cases with diabetes mellitus, mental disorders, and other immunodeficiencies (besides HIV) among cases of coinfection. There was also a decrease in the diagnosis of AIDS in the second study period, possibly resulting from efforts by the Ministry of Health to ensure implementation of the "treatment as prevention" strategy for all HIV cases, regardless of viral load and T-CD4 cell count [27]. These efforts aim to achieve the "95-95-95" goals, proposing to eliminate the global AIDS epidemic by 2030, with the testing of 95% of the HIV population, treatment of 95% of positive cases, and maintaining 95% of people in treatment with undetectable viral loads [28]. Brazil has achieved 88%, 83%, and 95% in these respective goals, but still

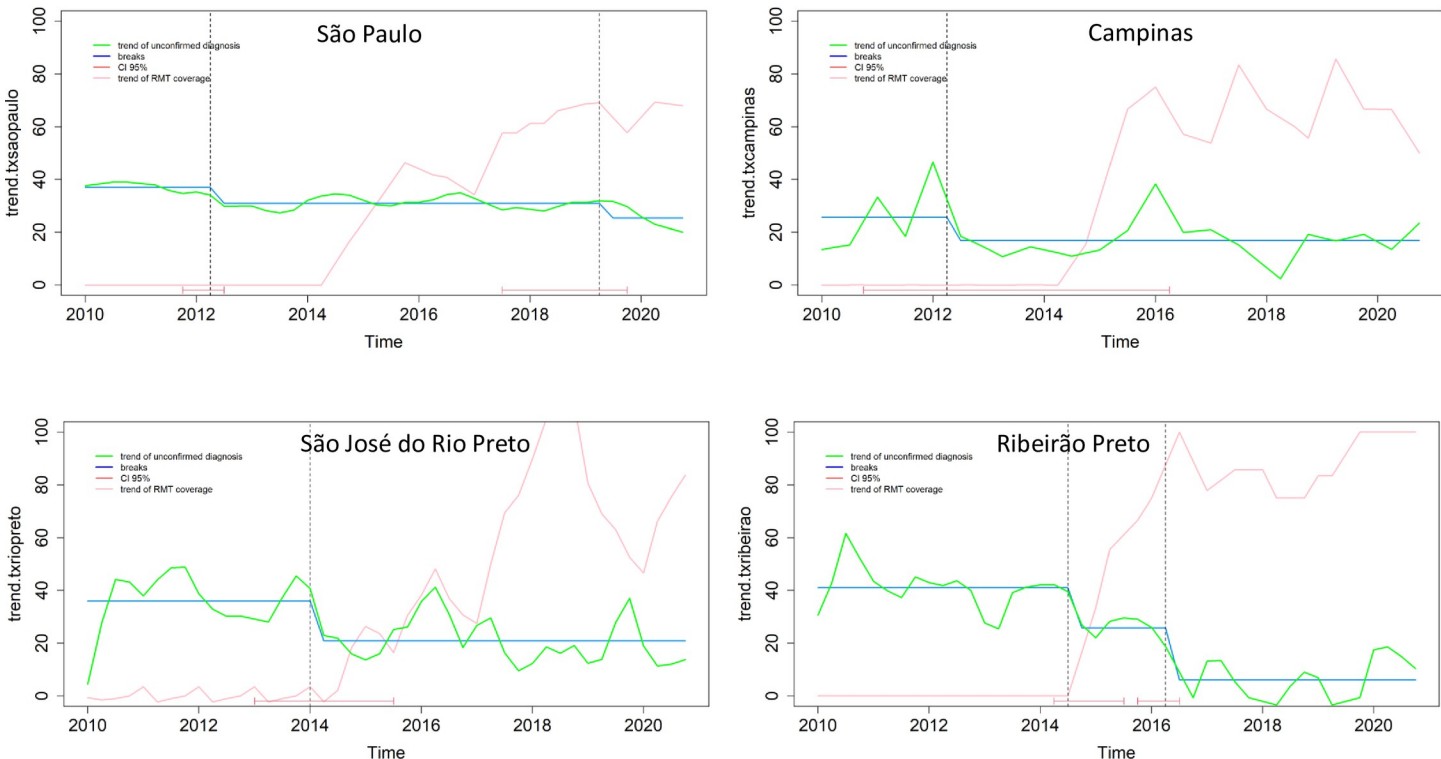

**Fig 3. Breaks and temporal trend of non-confirmed pulmonary tuberculosis among people with TB/HIV coinfection reported in four municipalities in the state of São Paulo between 2010 and 2020.**

**Table 3. Rates of non-confirmed pulmonary tuberculosis among people with TB/HIV coinfection in municipalities in the state of São Paulo, according to the implementation period of rapid molecular testing for tuberculosis (Xpert)* from 2010 to 2020.**

| Municipality | Pre-Xpert implementation (%) | Post-Xpert implementation (%) | p |
|---|---|---|---|
| Presidente Prudente | 44.4 | 37.8 | 0.520** |
| Bragança Paulista | 71.4 | 36.0 ↓ | 0.034** |
| Barueri | 25.0 | 31.6 | 0.544** |
| Taubaté | 52.6 | 31.5 ↓ | 0.042** |
| Franco da Rocha | 40.0 | 27.8 | 0.458** |
| Sorocaba | 28.3 | 27.5 | 0.907** |
| Carapicuíba | 35.7 | 24.3 | 0.317** |
| São Bernardo do Campo | 26.7 | 22.0 | 0.515** |
| Tremembé | - | 9.5 | 0.483*** |
| Marília | 10.2 | 6.3 | 0.344*** |
| Araçatuba | - | 6.3 | 0.762*** |
| Jundiai | 27.5 | 4.3 ↓ | 0.003** |
| Itapecerica da Serra | 25.0 | 3.1 ↓ | 0.025** |

*Each municipality had different dates for starting the Xpert implementation

**p-value for Chi-square Test

*** p-value for Fisher's Exact Test.

faces obstacles related to inequalities and difficulties in accessing HIV prevention and treatment actions [29].

Concerning the use of psychoactive substances, there was an increase in the use of legal and illegal drugs in the period after implementing Xpert among coinfection cases, indicating the need for integration between TB and HIV Control Programs and Psychosocial Care Centers for Alcohol and Drug Abuse, as well as the Smoking Control Program. This integration aims to screen cases treated in these services, increasing the detection of both infections and providing specialized and coordinated care to reduce or cease the use of psychoactive substances. These recommendations are valid, as the use of such substances is mentioned as a factor associated with TB development [30–32] and delays in seeking health services for TB diagnosis [33]. This delay is possibly explained by people's fear of developing withdrawal syndrome if diagnosed and requiring hospitalization, as well as the fact that TB symptoms such as cough, loss of appetite, and weight loss are ignored due to their similarity to the effects of drug use [34]. On the other hand, although it should be combatted, drug use seems to enhance detection of *Mycobacterium tuberculosis* by Xpert in sputum samples from PLHIV [35].

Although prevention and access to treatment are crucial for reducing the TB burden, the role of accurate and rapid diagnostic tests for disease control has gained greater visibility over the last decade. The RMT-TB performed by the GeneXpert system is considered a new test to improve TB diagnosis, as it has higher sensitivity than smear microscopy and provides results within two hours. It is also designed to diagnose rifampicin resistance. Being more specific, GeneXpert detects 40–60% more TB cases than smear microscopy when people with suspected TB are treated without microbiological confirmation [36, 37] and reduces the proportion of untreated patients with culture-positive TB from 40% to 15% [10].

Despite the potential of RMT-TB, this study shows that the test utilization rate in the second study period did not surpass the smear microscopy rate, reaching approximately 70% at the end of the period. This may have limited a more significant result regarding the test's effectiveness in reducing the rate of cases treated without etiological confirmation. In addition to not replacing smear microscopy, other reasons for underutilization of RMT-TB include inadequate logistics for sample delivery to laboratories, low active case finding, low sample quality, errors in filling out test requests, lack of human resources, workload overload, delay in equipment implementation, and equipment use only during one work shift [38].

When analyzing the time series of municipalities separately, there was heterogeneity in their performance regarding the impact of RMT-TB implementation on NC-PTB rates. It seems that this performance is mainly linked to the test utilization rate, as of five municipalities (Bauru, Guarulhos, São José do Rio Preto, São Paulo, and Santos) that had more than 20% of NC-PTB at the end of the study, four (Bauru, Guarulhos, São José do Rio Preto, and São Paulo) had less than 50% test utilization, while three municipalities (Santo André, Campinas, and Ribeirão Preto) with less than 20% of NC-PTB had more than 50% of RMT-TB utilization. Ribeirão Preto is noteworthy in this regard, as the test utilization trend in the municipality increased significantly after its implementation, and the time series shows two drops in the NC-PTB rate level. One may have occurred after the equipment implementation, and the other did occur after that event. This municipality has TB care centralized in five specialized outpatient clinics and the backup of a reference hospital for severe and challenging cases, which may have contributed to standardized practices and incorporating RMT-TB into care practices.

This points to the need for sensitization and continuous education of teams working in Primary Care services, including all their members, to prioritize RMT-TB requests when faced with a person suspected of TB [39].

In municipalities not analyzed by the time series, four of them (Bragança Paulista, Itapecerica da Serra, Taubaté, and Jundiaí) showed a decrease in the NC-PTB rate. Among these

municipalities, Jundiaí and Taubaté had high RMT-TB coverage (above 50%), while Bragança Paulista and Itapecerica da Serra presented the lowest coverage, showing that we need to better investigate what arrangements they did to improve their performance in confirming TB.

Eight municipalities (Presidente Prudente, Bragança Paulista, Taubaté, Barueri, Franco da Rocha, Sorocaba, Carapicuíba, and São Bernardo do Campo) still had a rate above 20% in the second study period (after Xpert implementation). Among these municipalities, six (Presidente Prudente, Bragança Paulista, Barueri, Franco da Rocha, Sorocaba, and Carapicuíba) had a test utilization rate below 50%.

Alland et al. (2015) [40] reported that the latest version of the test, GeneXpert Ultra, is more sensitive when compared to its initial version and likely as sensitive as liquid culture for TB diagnosis. Given the results of this study, it is possible to assume that the performance of RMT-TB is quite favorable for TB diagnosis in people with HIV, as stated by Weyer et al. (2013) [41], since they initiated TB treatment while waiting for the confirmation of TB results. Thus, Churchyard et al. (2014) [42] emphasized the use of RMT-TB as a substitute for smear microscopy in special populations, such as people living with HIV. This requires overcoming the challenges mentioned in this study for using technology as a tool for adequate and timely diagnosis of TB cases.

Therefore, services which act as the gateway to the health system must concentrate efforts on the effective and timely diagnosis of TB with Xpert ordering, aiming to reduce disease transmission, case worsening, and TB-related deaths [43].

As study limitations, it is worth noting a possible information bias due to the use of secondary data, as well as the difficulty of establishing a relationship between the exposure factor (implementation of RMT-TB) and the outcome (reduction in NC-PTB rates), which is inherent to an ecological study, as it involves the analysis of aggregated data, and therefore cannot be interpreted from an individual perspective.

## Conclusions

When PLHIV and suspected of PTB have access and are tested with XPERT in municipalities in the State of São Paulo, they have more confirmed diagnoses and less required therapy for TB. This also enables the physicians to know about the susceptibility profile of the *M. tuberculosis* for rifampicin. Despite the availability of Xpert test, suboptimal coverage in many sites may have affected the effect size on the trend reduction of NC-PTB in PLHIV.

These findings call attention and require more studies to understand why the State of São Paulo did not take the advantage of having the test for the population with suspected PTB and HIV, as a significant proportion of cases still do not have access to the molecular test. The best scenario was observed in Ribeirão Preto and Jundiaí, where there was high utilization of RMT-TB and a significant reduction in the NC-PTB rates in PLHIV. Centralized care of TB patients in specialized outpatient clinics, along with the presence of a reference hospital, appears to facilitate implementing standardized diagnosis, including integration of molecular testing into routine clinical practice.

## Supporting information

**S1 Data.**
(XLSX)

**S1 Data.**
(XLSX)

## Author Contributions

**Conceptualization:** Mariana Gaspar Botelho Funari de Faria, Rubia Laine de Paula Andrade, Valdes Roberto Bollela, Aline Aparecida Monroe.

**Data curation:** Mariana Gaspar Botelho Funari de Faria, Rubia Laine de Paula Andrade, Maria Josefa Perón Rujula, Aline Aparecida Monroe.

**Formal analysis:** Mariana Gaspar Botelho Funari de Faria, Rubia Laine de Paula Andrade, Antônio Carlos Vieira Ramos, Thais Zamboni Berra, Dulce Maria de Oliveira Gomes, Aline Aparecida Monroe.

**Funding acquisition:** Mariana Gaspar Botelho Funari de Faria, Rubia Laine de Paula Andrade, Aline Aparecida Monroe.

**Investigation:** Mariana Gaspar Botelho Funari de Faria, Rubia Laine de Paula Andrade, Aline Aparecida Monroe.

**Methodology:** Mariana Gaspar Botelho Funari de Faria, Rubia Laine de Paula Andrade, Dulce Maria de Oliveira Gomes, Valdes Roberto Bollela, Aline Aparecida Monroe.

**Project administration:** Mariana Gaspar Botelho Funari de Faria, Rubia Laine de Paula Andrade, Aline Aparecida Monroe.

**Resources:** Mariana Gaspar Botelho Funari de Faria, Rubia Laine de Paula Andrade, Aline Aparecida Monroe.

**Software:** Mariana Gaspar Botelho Funari de Faria, Rubia Laine de Paula Andrade, Aline Aparecida Monroe.

**Supervision:** Mariana Gaspar Botelho Funari de Faria, Rubia Laine de Paula Andrade, Aline Aparecida Monroe.

**Validation:** Mariana Gaspar Botelho Funari de Faria, Rubia Laine de Paula Andrade, Aline Aparecida Monroe.

**Visualization:** Mariana Gaspar Botelho Funari de Faria, Rubia Laine de Paula Andrade, Aline Aparecida Monroe.

**Writing – original draft:** Mariana Gaspar Botelho Funari de Faria, Rubia Laine de Paula Andrade, Karina Fonseca de Sousa Leite, Rafaele Oliveira Bonfim, Ana Beatriz Marques Valênça, Antônio Carlos Vieira Ramos, Thais Zamboni Berra, Ricardo Alexandre Arcêncio, Maria Josefa Perón Rujula, Jaqueline Garcia de Almeida Ballestero, Erica Chimara, Antônio Ruffino Netto, Dulce Maria de Oliveira Gomes, Valdes Roberto Bollela, Aline Aparecida Monroe.

**Writing – review & editing:** Mariana Gaspar Botelho Funari de Faria, Rubia Laine de Paula Andrade, Karina Fonseca de Sousa Leite, Rafaele Oliveira Bonfim, Ana Beatriz Marques Valênça, Antônio Carlos Vieira Ramos, Thais Zamboni Berra, Ricardo Alexandre Arcêncio, Maria Josefa Perón Rujula, Jaqueline Garcia de Almeida Ballestero, Erica Chimara, Antônio Ruffino Netto, Dulce Maria de Oliveira Gomes, Valdes Roberto Bollela, Aline Aparecida Monroe.

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
