## [Decision Letter · Decision Letter 0]

25 Mar 2024

PONE-D-23-42570Operational indicators for pulmonary tuberculosis diagnosis in people living with HIV before and after Xpert MTB/RIF implementation in the state of São Paulo, BrazilPLOS ONE

Dear Dr. Andrade,

Thank you for submitting your manuscript to PLOS ONE. After careful consideration, we feel that it has merit but does not fully meet PLOS ONE’s publication criteria as it currently stands. Therefore, we invite you to submit a revised version of the manuscript that addresses the points raised during the review process.

**I would like to sincerely apologise for the delay you have incurred with your submission. It has been exceptionally difficult to secure reviewers to evaluate your study. We have now received two completed reviews; the comments are available below. The reviewers have raised significant scientific concerns about the study that need to be addressed in a revision.**

**Please revise the manuscript to address all the reviewer's comments in a point-by-point response in order to ensure it is meeting the journal's publication criteria. Please note that the revised manuscript will need to undergo further review, we thus cannot at this point anticipate the outcome of the evaluation process.**

We look forward to receiving your revised manuscript.

Kind regards,

Miquel Vall-llosera Camps

Staff Editor

PLOS ONE

Journal Requirements:

3. Please provide additional details regarding participant consent. In the ethics statement in the Methods and online submission information, please ensure that you have specified (a) whether consent was informed and (b) what type you obtained (for instance, written or verbal, and if verbal, how it was documented and witnessed). If your study included minors, state whether you obtained consent from parents or guardians. If the need for consent was waived by the ethics committee, please include this information.

This study was financed by the Coordenação de Aperfeiçoamento de Pessoal de Nível Superior – Brasil (CAPES) [Financial Code 001], Conselho Nacional de Desenvolvimento Científico e Tecnológico (CNPq) – Produtivity in Research Sponsorship [grant number 317170-2021-0] and Fundação de Amparo à Pesquisa do Estado de São Paulo [process number 2022/00025-2].

This study was financed by the Coordenação de Aperfeiçoamento de Pessoal de Nível Superior – Brasil (CAPES) [Financial Code 001], Conselho Nacional de Desenvolvimento Científico e Tecnológico (CNPq) – Produtivity in Research Sponsorship [grant number 317170-2021-0] and Fundação de Amparo à Pesquisa do Estado de São Paulo [process number 2022/00025-2].

This study was financed by the Coordenação de Aperfeiçoamento de Pessoal de Nível Superior – Brasil (CAPES) [Financial Code 001], Conselho Nacional de Desenvolvimento Científico e Tecnológico (CNPq) – Produtivity in Research Sponsorship [grant number 317170-2021-0] and Fundação de Amparo à Pesquisa do Estado de São Paulo [process number 2022/00025-2].

7. For studies involving third-party data, we encourage authors to share any data specific to their analyses that they can legally distribute. PLOS recognizes, however, that authors may be using third-party data they do not have the rights to share. When third-party data cannot be publicly shared, authors must provide all information necessary for interested researchers to apply to gain access to the data. (https://journals.plos.org/plosone/s/data-availability#loc-acceptable-data-access-restrictions) 

a) A description of the data set and the third-party source

b) If applicable, verification of permission to use the data set

c) Confirmation of whether the authors received any special privileges in accessing the data that other researchers would not have

d) All necessary contact information others would need to apply to gain access to the data

Reviewers' comments:

Reviewer's Responses to Questions

**Comments to the Author**

1. Is the manuscript technically sound, and do the data support the conclusions?

Reviewer #1: Partly

Reviewer #2: Yes

2. Has the statistical analysis been performed appropriately and rigorously? 

Reviewer #1: No

Reviewer #2: No

3. Have the authors made all data underlying the findings in their manuscript fully available?

Reviewer #1: Yes

Reviewer #2: No

4. Is the manuscript presented in an intelligible fashion and written in standard English?

Reviewer #1: No

Reviewer #2: Yes

5. Review Comments to the Author

**Reviewer #1:** The study is very important because it is needed to improve the diagnosis of pulmonary TB in people living with HIV. However, the ecological design is not the type of study to conclude that people in the study need fewer treatments for TB. In addition, the authors compare the rates of unconfirmed diagnoses before and after the introduction of Xpert, but, they don't express this with a p-value (Table 3).

**Reviewer #2:** The manuscript by Faria et al. is an ecological study with time series analysis, investigating the impact of GeneXpert molecular testing implementation on confirmatory rates of pulmonary tuberculosis (PTB) among people living with HIV (PLHIV) across 21 municipalities in the State of São Paulo, Brazil, from 2010 to 2020. The findings demonstrate a decreasing trend in non-confirmed PTB rates among PLHIV, especially in locations with greater testing coverage.

Specific comments to the authors:

Minor comments:

1. In the Introduction section (lines 93-98), the authors mention a review manuscript conducted by their own group, without providing specific details on the overall conclusions drawn by this review, and without contextualizing the novelty of their current study. Additionally, the reference citation (Ref # 17) should be given in English, as there is a PDF source available and entitled “Effectiveness of GeneXpert® in the diagnosis of tuberculosis in people living with HIV/AIDS”.

2. It is recommended to provide clearer explanations in the Methods section regarding the molecular testing conducted with the GeneXpert® system. Specifically, clarify whether both versions of the test (Xpert® MTB/RIF and Xpert® Ultra) were utilized, and indicate whether the tests were conducted in a single reference laboratory or across multiple laboratories in the 21 municipalities. If the testing occurred in different laboratories, it is advisable to include details about the equipment used, particularly if the different equipment were all of the same model.

3. Concerning patient data, particularly alcohol and tobacco consumption (Lines 125-126), is there quantitative data available (e.g., for smoking, data on pack years)? It would be beneficial to include statistical analyses to assess the impact of smoking and alcohol drinking on comorbidity. Additionally, please reassess whether these variables are suitable to be considered as comorbidities in the context of the study, or if it would be more appropriate to include them as factors influencing comorbidities (such as diabetes and immunodeficiencies).

4. The authors should review grammar throughout the manuscript text, mainly in the Methods and Results sections. For example, in the Methods section, please review the grammar and conciseness of the information provided in Lines 129-135, as the text is not easy to follow. The same for the text presented in Lines 158-173.

5. Figure 1 (x-axis) label is written in Portuguese, and it should be translated to English.

6. Figure 2 legend will benefit from a more detailed explanation of the data.

7. In Lines 202-203, the authors state that, “Despite being available, not all PTB cases had a molecular test ordered and performed and the coverage of the exam varied among municipalities.” Please provide an explanation.

8. The authors should verify the text for typographical errors.

Major comments:

1. Did the authors adhere to a study guideline, such as STARD2015: An Updated List of Essential Items for Reporting Diagnostic Accuracy Studies, or any other pertinent guidelines for ecological time series studies in human health research? Please provide justification. If applicable, I suggest evaluating the study against suitable guidelines and citing the sources accordingly. While the authors have referenced (Ref # 18): Kleinbaum DG, Kupper LL, Morgenstern, H. Epidemiologic Research: Principles and Quantitative Methods. New York: John Wiley & Sons, 1982, it is advisable to incorporate an established and up-to-date guideline to ensure methodological rigor and transparent reporting of results.

2. In the Results section, Table 1 requires formatting. Furthermore, while the authors mention that they analyzed the study variables using descriptive statistics, it would be advantageous to include p-values for the data presented in this table. I recommend incorporating statistical analysis to determine whether a significant difference exists, given that the study's objective is to examine and compare non-confirmed pulmonary tuberculosis diagnoses before and after the implementation of GeneXpert® testing. If there are no statistically significant differences observed in any of the characteristics (before and after testing implementation), this should be clearly stated. Similarly, for Table 2, consider including p-values associated with the data for statistical analysis. Additionally, in the footnote of this table, it states: "Source: Authors, 2023." Please review this statement as it seems unnecessary.

3. In the Results section, the authors mention differences in ethnicity among individuals included in the study, particularly regarding characteristics such as TB/HIV coinfection (Lines 258-265). While it's understandable that this is a description of the data, it's important for the authors to ensure the use of appropriate language when reporting ethnicity. I recommend consulting the following guideline for guidance: Flanagin A, Frey T, Christiansen SL; AMA Manual of Style Committee. Updated Guidance on the Reporting of Race and Ethnicity in Medical and Science Journals. JAMA. 2021 Aug 17;326(7):621-627. PMID: 34402850.

4. In the Conclusions section, the authors should consider to include that centralized care of TB patients in specialized outpatient clinics, along with the presence of a reference hospital, appears to facilitate the implementation of standardized diagnostic, treatment, and follow-up protocols, including the integration of molecular testing into routine clinical practice.

6. PLOS authors have the option to publish the peer review history of their article (what does this mean?). If published, this will include your full peer review and any attached files.

Reviewer #1: **Yes: **Ivette Valcárcel Valcárcel

Reviewer #2: No

---

## [Author Response · Author response to Decision Letter 0]

29 Apr 2024

Dear reviewers,

Thank you for giving us the opportunity to submit a revised draft of the manuscript “Operational indicators for pulmonary tuberculosis diagnosis in people living with HIV before and after Xpert MTB/RIF implementation in the state of São Paulo, Brazil” for publication in the Plos One. We appreciate the time and effort that you and the reviewers dedicated to providing feedback on our manuscript and are grateful for the insightful comments on and valuable improvements to our paper.

We have incorporated most of the suggestions made by the reviewers. Those changes are highlighted within the manuscript. Please see below, a point-by-point response to the reviewers’ comments and concerns.

Journal Requirements:

PONE-D-23-42570

1. OK

2.OK

3. The data were obtained after receiving acceptance by the Research Ethics Committee. As this is anonymous data, there was no need for a Informed Consent Form from the study sample. The term "anonymously" was included in the methods section, as well as indicating that the study did not address people under 18 years of age.

4,5,6. This study was financed by the Coordenação de Aperfeiçoamento de Pessoal de Nível Superior – Brasil (CAPES) [Financial Code 001] (PhD scholarship), Conselho Nacional de Desenvolvimento Científico e Tecnológico (CNPq) – Produtivity in Research Sponsorship [grant number 317170-2021-0] (advisor's scholarship) and Fundação de Amparo à Pesquisa do Estado de São Paulo [process number 2022/00025-2] (research fund to present results in scientific events).

7. A complementary file with the data was shared with the manuscript.

Reviewer’s comments:

01. Reviewer #1: We believe that the modifications made in the results session of this study gave greater solidity to the conclusions;

02. Reviewer #1 and 2: The analyzes were revised based on the comments presented by the reviewers, making them more robust

03. The authors make the data available

04. Reviewer #1: The text has been revised and presented appropriately

05. Reviewer #1- We believe that after the results revision, the conclusions became more suitable, which were also reviewed.

Minor comments:

1.The review’s conclusion was in the text: a literature review in December 2019 highlighted a higher accuracy of Xpert in confirming pulmonary TB (PTB) in PLHIV compared to smear microscopy, and showed similar performance to culture and much faster

2. Lines- 113-116 were corrected

3. As this is secondary data, it was not possible to obtain information on the amount of substance use. The appropriate term to name it was included in the table and text - use of psychoactive substances. 

4. Lines-131-135 and 158-172 were corrected.

5. Figura 1 was translated.

6. We did not insert any explanation in Figure 2 legend, cause we think they are mentioned in the text.

7. Between lines 303-307, we have already showed possible explanations for the underuse of RMT-TB for diagnosing tuberculosis.

8. The text was reviewed.

Major comments:

1. This study was developed based on the Strengthening the Reporting of Observational Studies in Epidemiology (STROBE) initiative checklist [20] and the suggested additions to the STROBE document for ecologic reporting [21].

2. p-value was included on tables 1, 2 and 3. 

3. The authors carried out the suggested reading regarding the recommendations on writing Race/Ethnicity, adjusting the text as requested. However, the “Brown” race is a Brazilian term, therefore, we did not find a better way to name it.

4.We agree with the suggestion and included it on conclusions.

6. No. We do not want to publish our full peer review.

---

## [Decision Letter · Decision Letter 1]

23 May 2024

Operational indicators for pulmonary tuberculosis diagnosis in people living with HIV before and after Xpert MTB/RIF implementation in the state of São Paulo, Brazil

PONE-D-23-42570R1

Dear Dr. Andrade,

We’re pleased to inform you that your manuscript has been judged scientifically suitable for publication and will be formally accepted for publication once it meets all outstanding technical requirements.

Kind regards,

Vinícius Silva Belo

Academic Editor

PLOS ONE

Additional Editor Comments (optional):

Reviewers' comments:

Reviewer's Responses to Questions

**Comments to the Author**

1. If the authors have adequately addressed your comments raised in a previous round of review and you feel that this manuscript is now acceptable for publication, you may indicate that here to bypass the “Comments to the Author” section, enter your conflict of interest statement in the “Confidential to Editor” section, and submit your "Accept" recommendation.

Reviewer #1: All comments have been addressed

2. Is the manuscript technically sound, and do the data support the conclusions?

Reviewer #1: Yes

3. Has the statistical analysis been performed appropriately and rigorously? 

Reviewer #1: Yes

4. Have the authors made all data underlying the findings in their manuscript fully available?

Reviewer #1: Yes

5. Is the manuscript presented in an intelligible fashion and written in standard English?

Reviewer #1: Yes

6. Review Comments to the Author

Reviewer #1: I consider that the original study titled Operational indicators for pulmonary tuberculosis diagnosis in people living with HIV before and after Xpert MTB/RIF implementation in the state of São Paulo, Brazil, by Dr. Rubia Laine de Paula Andrade, was corrected appropriately, describing methods, results, and conclusion accord to the reporting guidelines. And the results are important for the implementation of tuberculosis control policies.

7. PLOS authors have the option to publish the peer review history of their article (what does this mean?). If published, this will include your full peer review and any attached files.

Reviewer #1: No

---

## [Editor Report · Acceptance letter]

30 May 2024

PONE-D-23-42570R1 

PLOS ONE

Dear Dr. Andrade, 

I'm pleased to inform you that your manuscript has been deemed suitable for publication in PLOS ONE. Congratulations! Your manuscript is now being handed over to our production team.

Kind regards, 

on behalf of

Dr. Vinícius Silva Belo 

Academic Editor

PLOS ONE